# Estimation of Serial Interval and Reproduction Number to Quantify the Transmissibility of SARS-CoV-2 Omicron Variant in South Korea

**DOI:** 10.3390/v14030533

**Published:** 2022-03-04

**Authors:** Dasom Kim, Sheikh Taslim Ali, Sungchan Kim, Jisoo Jo, Jun-Sik Lim, Sunmi Lee, Sukhyun Ryu

**Affiliations:** 1Department of Preventive Medicine, Konyang University College of Medicine, Daejeon 35365, Korea; dsarie7@gmail.com (D.K.); jisu5680@naver.com (J.J.); 2World Health Organization Collaborating Centre for Infectious Disease Epidemiology and Control, School of Public Health, Li Ka Shing Faculty of Medicine, University of Hong Kong, Hong Kong, China; alist115@hku.hk; 3Laboratory of Data Discovery for Health Limited (D24H), Hong Kong Science Park, New Territories, Hong Kong, China; 4Department of Applied Mathematics, Kyung Hee University, Yongin 17104, Korea; sckim@khu.ac.kr (S.K.); sunmilee@khu.ac.kr (S.L.); 5Department of Nursing, Konyang University, Daejeon 35365, Korea; 6IHAP, Université de Toulouse, INRAE, ENVT, 31300 Toulouse, France; jun-sik.lim@envt.fr

**Keywords:** omicron, transmissibility, reproduction number, serial interval, coronavirus, SARS-CoV-2, outbreak

## Abstract

The omicron variant (B.1.1.529) of severe acute respiratory syndrome coronavirus 2 (SARS-CoV-2) was the predominant variant in South Korea from late January 2022. In this study, we aimed to report the early estimates of the serial interval distribution and reproduction number to quantify the transmissibility of the omicron variant in South Korea between 25 November 2021 and 31 December 2021. We analyzed 427 local omicron cases and reconstructed 73 transmission pairs. We used a maximum likelihood estimation to assess serial interval distribution from transmission pair data and reproduction numbers from 74 local cases in the first local outbreak. We estimated that the mean serial interval was 3.78 (standard deviation, 0.76) days, which was significantly shorter in child infectors (3.0 days) compared to adult infectors (5.0 days) (*p* < 0.01). We estimated the mean reproduction number was 1.72 (95% CrI, 1.60–1.85) for the omicron variant during the first local outbreak. Strict adherence to public health measures, particularly in children, should be in place to reduce the transmission risk of the highly transmissible omicron variant in the community.

## 1. Introduction

In South Korea, the severe acute respiratory syndrome coronavirus 2 (SARS-CoV-2) lineage B.1.1.529 (omicron variant) was first identified in an international traveler from Nigeria on 25 November 2021. As of 25 January 2022, the omicron variant had overtaken the delta variant (B.1.617.2) and become predominant in South Korea [1].

The omicron variant has mutations in the spike protein and SARS-CoV-2 receptor-binding domain, which contributes to its escape from neutralizing antibodies and increases its potential for transmission [2]. Epidemiological studies in the U.S. and Norway demonstrated the transmission of the omicron variant in fully vaccinated individuals against SARS-CoV-2 [3,4]. However, the transmission dynamics of the omicron variant and its intensity, including estimations of its serial interval distribution and reproduction number (R), remain unclear, and these estimations in other locations will be interesting.

In South Korea, the community transmission (local) associated with the omicron variant was first identified on 2 December 2021. In this study, we took the opportunity to analyze the case data associated with the omicron variant outbreak and estimated the serial interval and the reproduction number in South Korea.

## 2. Materials and Methods

Since 2 December 2021, for the first outbreak that was related to a church, extensive contact tracing and home quarantine for individuals who had any exposure to the confirmed case were conducted to identify further cases of the omicron variant. The individuals were isolated in their homes with mandatory testing using a RT-PCR test when symptoms developed. Furthermore, to detect asymptomatic cases, testing was conducted on the first and last days for all individuals during their 14-day home quarantine.

### 2.1. Source of Data

To identify cases of the omicron variant, public health authorities conducted whole-genome sequencing using respiratory samples obtained from epidemiologically linked individuals who tested positive for SARS-CoV-2 on a RT-PCR test [5]. We collected publicly reported data of patients infected with the omicron variant from the Korean public health authorities between 25 November 2021 and 31 December 2021 [6,7]. The data included information obtained by contact tracing of other reported cases of the omicron variant, demographic characteristics of the cases, date of symptom onset, and source of infection (https://github.com/gentryu/COVID-19omicron, accessed on 1 March 2022).

### 2.2. Serial Interval and Reproduction Number

The serial interval is the time between symptom onset in the infector and infectee in a transmission chain. To avoid right censoring bias, we excluded infectors identified after 20 December 2021. We computed the serial interval as the number of days. Because of the negative serial intervals, we added a value of 10 before fitting the parametric distribution and removed a value of 10 after obtaining the estimate. Then, we fitted four different parametric distributions (normal, shifted log-normal, shifted gamma, and shifted Weibull) using maximum likelihood estimation [8]. We also opted for the best-fitted model for serial interval distribution based on the Akaike information criterion (AIC). To identify the difference in the serial interval distribution on the infectors’ age, we estimated the age-specific serial interval distributions by stratifying the transmission pairs data for the infector’s age group into children (<10 years) and adults (≥20 years) as infectors aged 10–19 years were fewer in number and not included in the study. Furthermore, we analyzed the statistical differences of the mean serial interval between two infectors’ age groups by using a two-sided *t*-test.

The reproduction number (*R*) represents the mean number of secondary infections per primary case, and an *R* value above 1 indicates the epidemic is growing [9]. For newly emerging infectious diseases, the number of cases exponentially increases in the initial phase of an outbreak, and the observed value of the intrinsic growth rate (*r*) is often related to the *R* [10]. The r is estimated via a stochastic pure birth process known as the Yule–Furry process [11], and its likelihood is proportional to
$$\exp\left(-r\overset{T-1}{\underset{t=0}{\sum }}C\left(t\right)\right){\left(1-\exp\left(-r\right)\right)}^{C\left(t\right)-C\left(t-1\right)},$$
where $C\left(t\right)$ denotes the cumulative number of cases on day t.

*R* can be defined as follows:$$R=e^{rT_{c}-\left(1/2\right)r^{2}\sigma^{2}},$$
where $T_{c}$ is the mean generation time which follows normal distribution, defined as the mean time interval between the occurrence of successive infections in a transmission chain, and σ is the standard deviation [10]. As the infection process is unobserved, generation time distribution is often approximated with a clinical measure: serial interval distribution [10,12].

We also conducted an alternative analysis using the attack rate (AR), which is the number of cases of the omicron variant infection divided by the number of people at risk of infection. Based on the classical compartment model of infectious disease [13], the AR is linked to the reproduction number by
$$R=-\frac{log\left(\frac{1-AR}{S_{0}}\right)}{AR-\left(1-S_{0}\right)}$$
where $S_{0}$ is the initial percentage of the susceptible population. As the rate of vaccination completed in the district where the outbreak occurred was 61.8% [14], we assumed that the initial percentage of the susceptible population for the omicron variant was 40%. Furthermore, we assumed 411 individuals among 780 who attended the same afternoon worship with the laboratory-confirmed case-patient were the population at risk of infection [15].

## 3. Results

### 3.1. Description of First Community Outbreak Transmitted from Imported Cases

A woman (case-patient 1) and her husband (case-patient 2) traveled to Nigeria on 14 November 2021. On 19 November, she experienced a sore throat and resolved her symptom without treatment on 22 November [16]. On their flight to South Korea on 24 November, case-patient 2 developed a chill. Because they had received two doses of COVID-19 vaccination (mRNA-1273) before travel, they were exempt from quarantine upon entry according to Korean Quarantine Exemption Guidelines [17]. After their arrival at Incheon airport in South Korea, a man in his 30s (case-patient 3), who had not received COVID-19 vaccination, picked them up in his private vehicle after a handshake with case-patient 2. They all wore face masks and traveled together to the local COVID-19 screening center to conduct nasopharyngeal swabs, which were mandatory for Korean international travelers [17]. Case-patients 1 and 2 were diagnosed with SARS-CoV-2 infection on 25 November, and contact tracing was conducted. However, case-patient 3 was not identified as a contactee due to misinformation. Case-patient 3 had a chill and tested positive for SARS-CoV-2 infection on 29 November, which was identified as the omicron variant on 2 December. Before symptom onset, he attended church on 28 November with his family and a friend. Public health authorities ordered a 14-day home quarantine for the possibly exposed individuals to the laboratory-confirmed cases. Overall, 76 case-patients were identified during contact tracing and quarantine period associated with case-patient 3 (Figure 1).

### 3.2. Estimation of Serial Interval and Reproduction Number

Out of a total 427 cases, 18% (*n* = 76) of cases were linked to a church, which was the location of the first local outbreak in South Korea. Of these, 30% (*n* = 23) of cases had received two doses of the COVID-19 vaccine (Table 1).

We reconstructed 73 transmission pairs from the known onset dates for the infectors and infectees using a similar algorithm reported earlier for mainland China data [18,19]. We found the normal distribution was the best fit for serial interval distribution, estimated as a mean of 3.78 days (95% credible interval (CrI) 3.02–4.54), and the standard deviation was 3.33 days (95% CrI, 2.56–4.09) (Figure 2A). The mean estimates that fitted other distributions were comparable and are presented in Appendix A. We determined the mean estimate of the serial interval was significantly shorter for the child infectors (mean 3.0 days) than for the adult infectors (5.0 days) (*p* < 0.01) (Figure 2B).

The reproduction number during the first local outbreak was estimated to be 1.72 (95% CrI, 1.60–1.85). For the alternative analysis using AR, we estimated the mean R was 1.72 (95% CrI, 1.67–1.76).

## 4. Discussion

Countries, including Japan, China, and the U.S., where a large portion of the population has received two doses of the COVID-19 vaccine, reported a surge in COVID-19 cases associated with the omicron variant in January 2022. Although the rapid spread of the omicron variant has caused global concern, the transmission dynamics of the omicron variant are not yet clear. In our study, we identified the transmission characteristics of the omicron variant including its serial interval and the reproduction number using local cases in Korea during the early phase of the omicron wave, at which time 80% of the population had received two doses of the COVID-19 vaccine and active case-finding techniques with a strict test-trace-isolation strategy were in place [20,21].

For the description of the transmission of the imported case, the public health authorities reported that case-patient 3 had physical contact with the first imported case. However, the exact transmission route of the infection remained unclear because the primary route of SARS-CoV-2 infection is respiratory.

A recent Korean study of the omicron variant estimated the mean serial interval to be 2.8 days, which is shorter than our estimates [22]. This discrepancy is likely due to the truncated observation period and the small sample size in the previous study. 

We found the mean serial interval of the child infector was 2 days shorter than that of adult infectors. Susceptible individuals are likely to become infected with SARS-CoV-2 more quickly. As of December 2021 in South Korea, COVID-19 vaccination had not been recommended for children under the age of 10 years. Therefore, this finding is likely due to the different levels of COVID-19 vaccination uptake, which may have contributed to the different immunological responses in the two different age groups [23]. Furthermore, the cases in children are comparatively less severe, with many asymptomatic cases and a delay in viral load dynamics causing late illness symptoms [24], leading to shorter serial intervals. Furthermore, the differences in behavioral factors and symptom expression in different age groups during their home quarantine may have affected our findings [25].

Regarding the comparison of the serial interval between the omicron and delta variants in South Korea, our estimate of the omicron variant is similar to that of the delta variant (estimated mean of 3.3–3.5 days) [20,26].

To compare the different estimates of reproduction numbers, we conducted an alternative analysis using the classical epidemic model. The estimate was similar when estimated with the AR. A previous Korean study estimated that the *R* associated with the delta variant outbreak under the strict test-trace-isolation strategy during the period when the vaccination program had not yet been extended to the public was 0.99 (95% CrI, 0.79–1.25) [20], which suggested that the omicron variant has an enhanced transmission potential compared to that of the delta variant. This is in line with a previous study that demonstrated the omicron variant is likely to have substantially increased immune escape capability [27]. Furthermore, our finding is similar to that of a recent study that estimated the *R* of the omicron variant as 1.3 times larger higher than that of the delta variant [28]. However, the different levels of immunity in affected populations and implemented public health measures might contribute to the difference in the estimate [29]. Therefore, careful interpretation in the comparison of the estimates acquired from different times and locations is necessary.

The increased transmissibility of the omicron variant was also observed in countries with an extensive vaccination program [30]. Therefore, strict adherence to public health measures, COVID-19 vaccination uptake, including booster doses, and rapid case-finding under the test-trace-isolation strategy are required to reduce the transmission risk and mitigate the transmission of the omicron variant in the community.

The first limitation of this study is that the estimation was calculated using symptom-based public press releases from public health authorities rather than epidemiological reports. Thus, our estimates of serial interval could be biased. However, the early estimate of epidemiological characteristics of emerging infectious diseases is often conducted using publicly available data [8,18,21]. Second, in the early phase of this outbreak, a 14-day home quarantine was implemented for all individuals who had any risk of exposure to a confirmed case. Therefore, our estimate of the serial interval has avoided longer serial intervals that can be found in uncontrolled epidemics [8]. Furthermore, transmissibility might not be generalized for other locations where different control strategies of SARS-CoV-2 had been implemented. Third, we did not analyze the impact of COVID-19 vaccination on the transmission and severity of the case due to data limitations.

## 5. Conclusions

In conclusion, during the early omicron wave in South Korea, the variant was highly transmissible, even in communities with strict measures. Therefore, additional research in different settings is warranted to provide evidence for efficient policy decisions to implement public health measures against the transmission of the omicron variant.

## Figures and Tables

**Figure 1 viruses-14-00533-f001:**
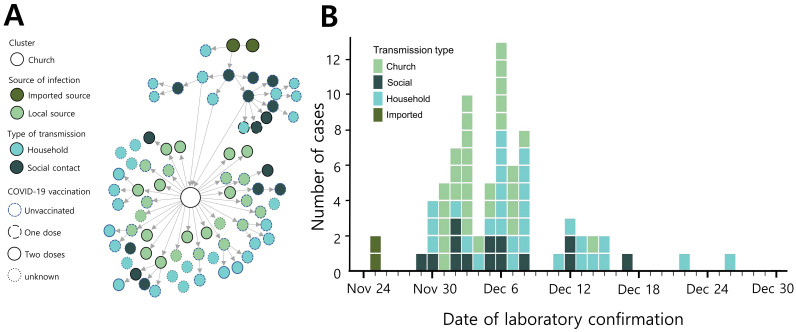
Transmission of the omicron variant of COVID-19 in South Korea in 2021. Transmission chain (**A**) and epidemic curve (**B**) of laboratory-confirmed SARS-CoV-2 Omicron variant infection associated with the first local outbreak (*n* = 76) in South Korea.

**Figure 2 viruses-14-00533-f002:**
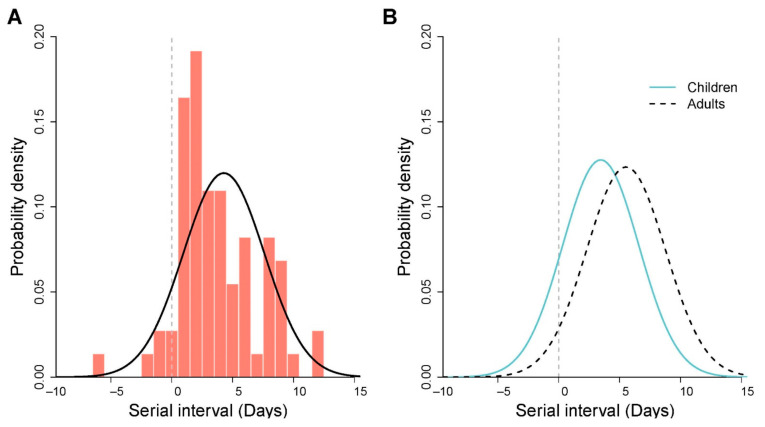
Estimated serial interval distributions of the SARS-CoV-2 omicron variant in South Korea. (**A**) The estimated serial interval distribution was analyzed using the 73 infector–infectee pairs. The vertical red bars indicate the empirical density of the serial intervals calculated from transmission pair data, and the solid black curve indicates fitted normal distribution. Gray vertical lines indicate the presymptomatic transmissions. (**B**) The estimated serial interval distributions were analyzed by two different age groups of infectors. The fitted serial interval distributions (normal) for child infectors (*n* = 44 pairs) in solid blue curve and adult infectors (*n* = 29 pairs) in dashed black curve.

**Table 1 viruses-14-00533-t001:** Descriptive summary of the demographic and epidemiological characteristics of 427 confirmed cases of the SARS-CoV-2 omicron variant in South Korea.

Overall	Total (%)	Church-Related	Other Settings ^†^
	427	76	351
Age group (years)			
0–9	93 (21.8%)	9 (11.8%)	84 (23.9%)
10–19	12 (2.8%)	5 (6.6%)	7 (2.0%)
20–59	123 (28.8%)	43 (56.6%)	80 (22.8%)
Above 60	20(4.7%)	6 (7.9%)	14 (4.0%)
Unknown	179 (41.9%)	13 (17.1%)	166 (47.3%)
COVID-19 vaccination			
None	84 (19.7%)	38 (50.0%)	46 (13.1%)
One dose	2 (0.5%)	2 (2.6%)	0
Two doses	70 (16.4%)	23 (30.3%)	47 (13.4%)
Unknown	271 (63.5%)	13 (17.1%)	258 (73.5%)
Type of transmission			
Imported	5 (1.2%)	2 (2.6%)	3 (0.9%)
Church	26 (6.1%)	26 (34.2%)	0
Household	110 (25.8%)	33 (43.4%)	77 (21.9%)
Social contact	285 (66.7%)	15 (19.7%)	270 (76.9%)
Unknown	1 (0.2%)	0	1 (0.3%)

^†^ Other settings include kindergartens, workplaces, and restaurants.

## Data Availability

The data generated during and/or analyzed during the current study are available on the open repository https://github.com/gentryu/COVID-19omicron (accessed on 1 March 2022).

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
