# Peer review of "Estimation of Serial Interval and Reproduction Number to Quantify the Transmissibility of SARS-CoV-2 Omicron Variant in South Korea"

_viruses, 2022, doi:10.3390/v14030533_

Round 1

Reviewer 1 Report

Article is well written no major changes needed .Difference between severity between Delta and Omicron wave needs to be discussed. 

Reviewer 2 Report

The authors present a brief report on estimation of the serial interval for Omicron variant during the early stage of the associated epidemic in South Korea. Although, the results are of interest for epidemiologists, the estimation was done using public press-releases of KCDC. The authors do not outline and discuss the limitations of this study design. For example, how definitive that selected epidemiologically linked pairs are actual infector-infectee pairs? Second, the main source of infections was a likely superspreading event in a church, however, the authors do not provide any descriptive information about it. Was there a definitive index case for it? Or people attended the church service several times? The authors reconstruct the transmission chain (Figure 1), but they do not give any information about the way they reconstruct it.

After reading the manuscript, I believe that a lot of details of both materials and applied methods are lacking and require more work from the authors. In my opinion, they need to provide more details and make the study more rigorous before considering it for publication. I would recommend major revision if the authors are willing to improve their manuscript.

Remarks:

 - Abstract, first sentence: "Omicron [...] is now the predominant variant". What do the authors mean by "now"? For example, what if a reader would check their paper in 2025? It is also good to avoid the phrase "now overtaken" as in the Introduction

 - Abstract: Serial interval has two ends, and the child/adult can be either infector or infectee. The authors need to clarify what they mean by SI among children and among adults as they did later in the main text (by infector's age). Also, their definition of a children's age is too broad (<20 y.o.) and is needed to mentioned.

 - Introduction: I think that the introduction should provide more details because the submitted work is for Original research rather than Short communications. For example, when did the government of South Korea implemented extensive contact tracing and active case finding? Was it done since 2 December 2021? Were the close contacts quarantined during that period? For example, Ali et al Science showed that this might affect the SI and make it shorter.

 - Methods: I would definitely not call 19y.o. as children. I think the contact pattern could be very different for high school compared to kindergarten as well as immunity response.

 - L72: I think the authors mean that the generation time follows gamma distribution in their Eq.1. In my opinion, this can be written rather than denoted by a general phrase "based on the epidemic theory..."

 - Supplementary Materials: given that the main text is short, the SMs can be moved completely to the main text as the Materials and Methods section.

 - Supplementary Materials: "In South Korea, as of early December 2021, 80% of the population have received full COVID-19 vaccination." - What's about waning efficiency of the vaccine and possible vaccine escape by Omicron? I think it is too quick to say that if 80% were vaccinated, 80% were protected. The vaccination among the church was much lower (38%), would this not affect the conclusions?

 - Table S2: Log-normal, gamma, and Weibull have positive support and do not accept negative SIs, does it mean that the authors did not identify any negative SIs in their dataset?

Round 2

Reviewer 2 Report

Thanks to the authors for their detailed response. I still have some critique - in my opinion, some details need to be clarified and corrected. The authors also write about using Bayesian framework but apply the Akaike information criterion (AIC) which can be used only for frequentist approach. This is wrong practice and is needed to be corrected. The authors need to use WAIC or LOOIC which are well-described in the literature. Otherwise, they need to switch to MLE and use AIC.

Regarding their response about using news media as a source of data, this should be reflected in the text. Rather than writing that "a similar algorithm was used", they can specify that the data collected from the media were used for the analysis. Given non-specificity of symptoms of Omicron and that the authors didn't have access to epidemiological reports, this can be regarded as limitation of the study and needed to be mentioned. For example, Du et al [9] specifically state about that in their paper.

Remarks: 
- L175 and L177: "14 days of home isolation" - I don't understand why isolation? Those are close contacts, and presumably not yet confirmed to be infected. So, it should be home quarantine, not isolation? See definition https://www.cdc.gov/quarantine/index.html
- L177: Provided reference [5] does not state that 411 individuals were quarantined. Namely, I read: "After the initial patients participated in a church program for foreigners on Nov. 28 attended by 411 people, infections spread to other church members, their families and acquaintances. Officials believe a total of 780 people visited the church on Nov. 28." - it says that 411 individuals attended the program, but overall 780 people visited the church. Why do the authors think that those 780 people were not quarantined? It definitely does not mean that 411 were quarantined - they could be advised to do health monitoring. The statement of the authors should be rephrased or supported by another source. 
- L194: "We computed the serial interval as the number of days and added 194 value of 10 before fitting parametric distribution and removed the value after having the 195 estimate to accept the negative serial intervals for the estimate." - Why did the authors do that? It is unclear from this sentence. I would recommend to rephrase it. I understand that it is because of negative SI, but why not to write about it?
- L196-199: "Then, we fitted four different parametric distributions (normal, shifted log-normal, shifted gamma, and shifted Weibull) using a Bayesian inferential framework to estimate [9]. We also opted for the 198 best-fitted model for serial interval distribution based on the Akaike information criterion (AIC)." - Here the authors mix up different things. AIC is for point estimates given by maximum likelihood estimation (frequentist approach), but it is not applicable to Bayesian stats. So the methodology to compare the models used by the authors is improper. They should use, for example, either WAIC or LOOIC. Second, the used reference of Du et al [9] applied maximum likelihood estimation and then AIC for model selection, which was Ok. However, citation on [9] that it used Bayesian framework and shifted distributions is not very correct. Du et al did not use them both.
- L301-303: "After their arrival at Incheon airport in South Korea, a 301 man in 30s (case-patient 3), who did not receive COVID-19 vaccination, picked them up 302 in his private vehicle after a handshake with case-patient 2." - It is unclear that the handshake was a causal link to infection of case-patient 3. I would say it is unproven. Otherwise, there should be some investigation and some proof that case-patient 3 touched their nose/mouth through the mask while driving or later. But it could be also respiratory transmission while driving (maybe the driver was wearing the mask improperly during the ride). The authors need to provide more details or reframe the statement. Definitely, they need to be move careful.
- L395-397: the authors need to specify if the difference in the means was *significant* or *clearly seen*. Both reported values are the means of the posteriors, but their uncertainty intervals can be overlapping. This is especially important because it is unclear if the group of infectors <10y.o. was sufficiently large for proper assessment of difference in the means of SI.
- L395-397: I am also worried that the children were unvaccinated, so it is unclear if the shorter SI could be caused by vaccination status rather than by age of the infector. It would be nice also to discuss this aspect if possible.

Typos, etc.
- L178: I think the authors need to be careful and write more precisely. RT-PCR test? not just RT-PCR.
- L200-203: "To identify the impact..." the sentence is not very clear - I recommend to revise it.
- Ref. 12: please provide the chapter and its authors, rather than the editors of the handbook. 
- L297: What does it mean "she [...] self-limited on 22 November"? Please correct the sentence.

Round 3

Reviewer 2 Report

I think it looks good now. I have no further remarks except one.

- Even though the authors removed their sentence about hand-shake between case patient 2 and case patient 3 according to my recommendation, I think they could still mention it, but to add that the exact transmission route of the infection remained unclear because SARS-CoV-2 primary route is respiratory. I apologies for some misunderstanding in that remark.
